# Long Non-Coding RNA Expression in Laser Micro-Dissected Luminal A and Triple Negative Breast Cancer Tissue Samples—A Pilot Study

**DOI:** 10.3390/medicina57040371

**Published:** 2021-04-12

**Authors:** Anca Marcu, Diana Nitusca, Adrian Vaduva, Flavia Baderca, Natalia Cireap, Dorina Coricovac, Cristina Adriana Dehelean, Edward Seclaman, Razvan Ilina, Catalin Marian

**Affiliations:** 1Department of Biochemistry and Pharmacology, Victor Babeş University of Medicine and Pharmacy, Pta Eftimie Murgu Nr.2, 300041 Timişoara, Romania; marcu.anca@umft.ro (A.M.); nitusca.diana@umft.ro (D.N.); eseclaman@umft.ro (E.S.); cmarian@umft.ro (C.M.); 2Department of Microscopic Morphology, Victor Babeş University of Medicine and Pharmacy, Pta Eftimie Murgu Nr.2, 300041 Timişoara, Romania; vaduva.adrian@umft.ro (A.V.); baderca.flavia@umft.ro (F.B.); 3Department of Pathology, Emergency City Hospital, 300041 Timişoara, Romania; 4Department of Surgical Oncology, Victor Babeş University of Medicine and Pharmacy, Pta Eftimie Murgu Nr.2, 300041 Timişoara, Romania; nata_cireap@yahoo.com; 5Department of Surgical Oncology, Municipal Hospital, Str. Gheorghe Dima Nr.5, 300254 Timişoara, Romania; 6Faculty of Pharmacy, Victor Babeş University of Medicine and Pharmacy, Pta Eftimie Murgu Nr. 2, 300041 Timişoara, Romania; dorinacoricovac@umft.ro (D.C.); cadehelean@umft.ro (C.A.D.)

**Keywords:** breast cancer, tissue specificity, lncRNA, laser capture microdissection

## Abstract

*Background and Objectives:* Breast cancer (BC) remains one of the major causes of cancer death in women worldwide. The difficulties in assessing the deep molecular mechanisms involved in this pathology arise from its high complexity and diverse tissue subtypes. Long non-coding RNAs (lncRNAs) were shown to have great tissue specificity, being differentially expressed within the BC tissue subtypes. *Materials and Methods:* Herein, we performed lncRNA profiling by PCR array in triple negative breast cancer (TNBC) and luminal A tissue samples from 18 BC patients (nine TNBC and nine luminal A), followed by individual validation in BC tissue and cell lines. Tissue samples were previously archived in formalin-fixed paraffin-embedded (FFPE) samples, and the areas of interest were dissected using laser capture microdissection (LCM) technology. *Results:* Two lncRNAs (OTX2-AS1 and SOX2OT) were differentially expressed in the profiling analysis (fold change of 205.22 and 0.02, respectively, *p* < 0.05 in both cases); however, they did not reach statistical significance in the individual validation measurement (*p* > 0.05) when analyzed with specific individual assays. In addition, GAS5 and NEAT1 lncRNAs were individually assessed as they were previously described in the literature as being associated with BC. GAS5 was significantly downregulated in both TNBC tissues and cell lines compared to luminal A samples, while NEAT1 was significantly downregulated only in TNBC cells vs. luminal A. *Conclusions:* Therefore, we identified GAS5 lncRNA as having a differential expression in TNBC tissues and cells compared to luminal A, with possible implications in the molecular mechanisms of the TNBC subtype. This proof of principle study also suggests that LCM could be a useful technique for limiting the sample heterogeneity for lncRNA gene expression analysis in BC FFPE tissues. Future studies of larger cohort sizes are needed in order to assess the biomarker potential of lncRNA GAS5 in BC.

## 1. Introduction

Breast cancer (BC), a major health concern, is considered to be the second leading cause of cancer death in women worldwide, with an estimate of 276,480 new cases in 2020 in the US alone, accounting for an estimate of 42,170 new deaths [1]. Heterogenous and complex by nature, BC comprises various and distinct patterns, which are classified in groups and subgroups that aid in the treatment standardization and patient care [2].

There are diverse means of classifying BC. A less costly approach provides an approximate classification based on immunohistochemistry (IHC) [3,4]. However, based on the genes expressed in BC tissues, there are five distinct subtypes: Luminal A, Luminal B, Triple negative (TNBC)/basal-like, HER2-enriched, and Normal-like, each having different patterns and particularities [5].

In the last decades, abundant literature findings indicate a novel approach toward the analysis of nucleic acids and protein metabolites, which could identify such molecules in archived tissue samples [6]. The high-throughput transcriptome data from BC patients indicated mounting evidence of long non-coding RNA (lncRNA) species that are differentially expressed in tumor tissue samples compared to normal ones [7].

LncRNAs are very abundant and diverse RNA transcripts, with lengths of over 200 nucleotides, that are not generally translated into proteins, and that possess various functions, from which some of the biological mechanisms remain to be understood. However, it is now known that there are several types of lncRNA, such as intergenic-, intronic-, bidirectional-, overlapping sense-, antisense lncRNAs, as well as lncRNAs hosted by a microRNA gene/cluster, with roles at different levels (transcriptional, posttranscriptional, and epigenetic). Moreover, reports showed that lncRNAs are involved in various pathophysiological processes, such as cell cycle regulation, chromatin remodeling, histone modifications, as well as gene imprinting, while others were found to interfere with transcription [8]. In cancer, studies showed that lncRNAs interact with several signaling pathways (Akt-, MAP kinase-, Wnt-, MYC signaling pathways), thus mediating cell proliferation, invasion, metastasis, and apoptosis [9].

In BC in particular, lncRNAs tend to reveal a tissue-specific pattern, as they are differentially expressed in the various subtypes of BC tissues. Microarray findings showed a dysregulated lncRNA expression profile in TNBC tissues, which lead to the hypothesis that they may be involved in the progression of this particular type of BC [10]. Furthermore, a wide number of studies indicated a possible link between lncRNAs involved in different molecular subtypes of this malignancy: LOC554202 was firstly discovered to play a role in the aggressive, estrogen receptor negative TNBC subtype [11], while LOC100288637 had the highest correlation with the HER2 positive subtype [12].

However, available literature reports that investigate comparative differential expression of lncRNAs in distinct tissue subtypes of BC are limited, since the majority of study designs include healthy subjects or adjacent non-tumor tissues as control samples [7]. Moreover, researchers have mainly studied the involvement of lncRNAs in BC using either fresh tissue specimens, TNBC cell lines, or archived FFPE tissue samples using regular protocols. As it is well known that BC is very heterogenous even within the same tissue slide, therefore, we attempted to limit this heterogeneity from our samples by using laser capture microdissection (LCM) for the dissection of our archived formalin-fixed paraffin-embedded (FFPE) specimens. LCM shows to represent a useful technique for BC tissue analysis, as it is capable of isolating desired cell populations from a very diverse tissue slide [13].

Therefore, we analyzed herein the lncRNA differential expression in two BC subtypes, luminal A and TNBC, in LCM-dissected FFPE archived tissue samples, so as to investigate not only information about the potential BC tissue specificity of different lncRNA species as biomarkers for BC but also for the technological aim of proving that LCM could be a powerful technique for isolating tumor tissues of interest for downstream gene expression analysis. Our study design involved a two-step approach: the “profiling” step, which included the analysis of a large lncRNA panel, and the “validation” step, where we analyzed individual lncRNAs, using specific primer sequences for those regions.


## 2. Materials and Methods

### 2.1. Patients’ Characteristics and Tumor Samples

The full description of the study design and patients’ characteristics can be found in two of our previous publications [13,14]. To summarize, we analyzed 18 breast tumor slides from patients with diagnosed invasive ductal breast carcinoma, who underwent surgical resection at the Department of Surgical Oncology of the Timişoara Municipal Hospital over one year period (2009–2010). We obtained informed consent from all patients before surgery for using their FFPE tumor tissue slides from the Pathology Department’s archive. The study was approved by the Ethical Committee of our institution and performed in accordance with the Ethical Standards of the 1964 Declaration of Helsinki and its later amendments.

### 2.2. Laser Capture Microdissection (LCM)

The preserved tumor tissues were cut with a MMI SmartCut Plus System (MMI Molecular Machines & Industries, Glattbrugg, Switzerland), isolating from the tissue samples only the malignant groups of cells. This method was described as being able to isolate, procure, and analyze desired cell populations, by cutting a specific area of a tissue sample under microscopic visualization. Despite its limitations, LCM proved to be suitable in the application of RNA transcript profiling, as previously described [13].

Briefly, the FFPE samples were sectioned with a microtome at 10 µm, mounted on slides free of RNase (MMI MembranSlides, MMI, Glattbrugg, Switzerland), and cut with adequate power and focus for UV laser shots with the aforementioned MMI SmartCut Plus System. The areas were captured, pooled, and placed in a microcentrifuge tube (RNA-free), as indicated by the manufacturer (mmi IsolationCap tubes, MMI, Glattbrugg, Switzerland). Typical slide images and a detailed protocol are described in two of our previous publications [15,16].

### 2.3. RNA Extraction

After microdissection, the cut regions were placed into a collection vessel and underwent a deparaffinization step. The deparaffinization technique used the melting protocol due to the small quantity of the samples. Paraffin was melted and cooled, and the solid layer was pierced with a tip, as indicated by the manufacturer for the miRNeasy FFPE kit (Qiagen, Hilden, Germany).


Total RNA was extracted using the same miRNeasy FFPE kit (Qiagen, Germany) according to the manufacturer’s instructions, using the spin column technique for RNA purification, which included ethanol precipitation of nucleic acids.


Following extraction, we eluted the samples with 12 μL RNase-free water and quantified the RNA concentration with the Qubit RNA HS Assay Kit (ThermoFisher Scientific, Waltham, MA, USA) using a Qubit^®^ Fluorometer.

### 2.4. LncRNA Profiling in FFPE Tissues

Following total RNA extraction, we performed reverse transcription reactions in order to obtain cDNA for the qRT-PCR amplification step. A fixed volume of 8 μL from the total RNA was input into subsequent cDNA reactions, using the RT2 PreAMP cDNA Synthesis kit (Qiagen, Germany). This includes a preamplification step in order to improve the low copy number lncRNA detection. The cycle conditions for the preamplification of cDNA from our FFPE tissue samples were selected according to the manufacturer’s indications with a 10 min (95 °C) HotStart DNA *Taq* Polymerase activation step followed by 8 cycles (95 °C/15 s, 60 °C/2 min). Quantitative real-time PCR was immediately performed to quantify 84 lncRNAs using the RT^2^ lncRNA PCR Array Human lncFinder (LAHS-001ZE; Qiagen, Germany) combined with RT^2^ SYBR^®^ Green qPCR Mastermix (Qiagen, Germany). Each reaction was performed in triplicate, with the following cycle conditions indicated by the protocol: 95 °C/10 min (1 cycle), 95 °C/15 s, 60 °C/1 min (40 cycles).

### 2.5. LncRNA Validation

The differentially expressed lncRNAs (OTX2-AS1 and SOX2OT), together with GAS5 and NEAT1 selected from literature were further validated using TaqMan Fast Advanced Master Mix/TaqMan individual assays, (assay ID Hs01008264_s1 for NEAT1, and Hs03464472_m1 for GAS5, respectively, from Thermo Fisher Scientific, USA), on luminal A (*n* = 9) and TNBC tissue samples (*n* = 9). ACTB was used as a housekeeping gene for data normalization, both in the profiling and validation experiments. RNA extraction and cDNA synthesis were performed using the same kits as previously described, and cDNA was subsequently used for qPCR reactions in a 7900 HT Real-Time PCR System (Thermo Fisher Scientific, USA).

### 2.6. Cell Lines

Two BC cell lines representative for the two molecular subtypes were cultured under standard conditions. Mda-mb-231 (TNBC) and MCF7 (LuminalA) cells were collected upon confluence, and the individual expression of GAS5 and NEAT1 was measured as described above.

### 2.7. Statistical Analysis

The lncRNA expression level was calculated using the comparative 2^−∆∆CT^ method, relative to the selected controls [17]. Delta CT was calculated between the gene of interest and the reference gene ACTB. The CT cut-off was set to 37.

We used Student’s t-test to compare the relative lncRNA expression between the control group (Luminal A) and tested group (TNBC). The Mann–Whitney nonparametric test was used for cell line comparisons, given the small number of samples in each group. The cutoff value for statistical significance was set at *p* < 0.05.


## 3. Results

The demographic characteristics of the patients, as well as the pathological features of the tumors are described in Table 1. The data are shown separately for TNBC and Luminal A groups.


The vast majority of the patients were older than 50 years in both groups (77.78% and 88.89% in the TNBC and Luminal A group, respectively). More than half of the patients (55.56%) had normal BMI values in both groups. Regarding the tumor characteristics, the patients presented in general tumor sizes smaller than 5 cm (55.56% for the TNBC group and 66.67% for the Luminal A group), from which more than half (55.56%) were HER2 positive in both groups.


Table 2 shows the determined lncRNA species from the profiling step, with their corresponding fold changes and *p* values. Only OTX2-AS1 and SOX2-OT had significant differences in gene expression. OTX2-AS1 had a fold change of 205.22 (*p* = 0.029), being overexpressed in TNBC tissues compared to luminal A, while SOX2-OT was downregulated, having a fold change of 0.02 (*p* = 0.042). In the validation step, none of these two lncRNAs presented statistical differences in gene expression (data not shown).


Although GAS5 and NEAT1 did not reach statistical significance in the profiling step, we performed an individual validation for these two lncRNAs as they are highly associated with BC. GAS5 showed to be significantly downregulated in TNBC tissue samples compared to Luminal A tissue samples, and it was also downregulated in Mda-mb-231 cell line compared to MCF7 cells. NEAT1 was also downregulated in both sample types of TNBC versus Luminal A; however, the comparison reached statistical significance only for cell lines, as shown in Table 3 and Figure 1a–d, respectively.

## 4. Discussion

To our knowledge, this is the first study that assesses the differential expression of GAS5 lncRNA in different BC tissue subtypes using LCM technology. Therefore, we proved that LCM is a powerful tool that is capable of isolating specific, tumor-enriched regions of interest from archived (FFPE) tissue samples for a more accurate lncRNA profiling. The aim of our study was also to investigate potential differences in the expression levels of lncRNAs in two different tissue subtypes (TNBC and luminal A), as tissue-specific biomarkers for BC.

Our findings showed that two lncRNAs (OTX2-AS1 and SOX2-OT) were differentially expressed (*p* < 0.05) in TNBC tissues compared to luminal A; however, they were not subsequently validated. Previous literature studies demonstrated that the pluripotency-associated transcription factor SOX2 gene is expressed and associated with at least 25 types of cancer, including BC [18]. Amaral et al. (2009) showed that the SOX2 gene is found in an intron of the multi-exon SOX2-OT lncRNA, and therefore, SOX2-OT acts as a transcriptional enhancer for the SOX2 gene [19].

However, little is known about the role of OTX2-AS1 in BC. In our study, OTX2-AS1 was also differentially expressed (*p* < 0.05) in TNBC tissues when compared to Luminal A samples in the profiling analysis.

In addition, as lncRNA clustering with mRNA PAM50 classification is tightly correlated [20,21], our study underwent in the same direction, proving that some lncRNA species are differentially expressed in TNBC tissues when compared to luminal A tissue samples.

Furthermore, we selected for validation two other individual lncRNAs that were previously associated with BC, the growth arrest-specific transcript 5 (GAS5) and the nuclear enriched abundant transcript 1 (NEAT1), as they are two of the most studied lncRNAs in BC [22,23,24,25,26]. A study conducted by Li et al. (2018) reported that GAS5 levels are decreased in TNBC types, showing its potential role as a tumor suppressor. The ectopic expression of GAS5 in TNBC tissue samples showed an enhanced apoptosis and reduced proliferation of TNBC cells [22]. In order to enhance the tumor suppressor role of GAS5, Pickard et al. (2016) showed that GAS5 oligonucleotides could serve as therapeutic targets in BC, as the hormone response element mimic (HREM) sequence within GAS5 alone can promote apoptosis of BC cells in the same manner as does the lncRNA GAS5 in full-length [27]. Our study corroborated with previous literature findings, GAS5 being significantly downregulated in TNBC tissues compared to luminal A tissues, as well as in the Mda-mb-231 cell line compared to MCF7 cells. The downregulation of GAS5 was correlated with a more aggressive phenotype [22], as TNBC subtype is hormone-receptor and HER2 negative and frequently associated with BRCA1 gene mutations, while luminal A is hormone-receptor positive, HER2 negative with low protein Ki-67 levels, thus having the best prognosis [5].

NEAT1 was also downregulated in TNBC cases in our study, although it did not reach statistical significance in tissue samples. Several reports have linked the abnormal expression of NEAT1 with BC, suggesting that the expression level of NEAT1 was significantly upregulated (6.86-fold increase) in TNBC tissues when compared to normal controls. The same report conducted by Shin et al. (2019) showed that NEAT1 promoted drug resistance and cancer stemness, and that NEAT1 knockdown minimized cell growth in cisplatin- or taxol-treated cells. NEAT1 silencing also decreased the tumor growth in vivo, suggesting therefore its oncogenic role [28]. Moreover, it was shown that NEAT1 is a direct transcriptional target of p53, and that the low expression of lncRNA NEAT1 in TNBC tissues is associated with poor prognosis of BC [29].

NEAT1 was also found to be significantly (*p* = 0.035) downregulated in our cell lines study in TNBC cells versus Luminal A, which is inconsistent with other literature studies. Previously, it was found that NEAT1 expression level is significantly upregulated in BC cell lines when compared to normal MCF-10A cells [30]. However, in our study, we used Luminal A cells (MCF7) as controls. Another report found NEAT1 levels as being upregulated in BC cells compared to normal cells, and that NEAT1 levels are inversely correlated with miR-133b, which in turn promotes BC migration and invasion [31].

Taken together, we suggest that GAS5 could be used as a tissue-specific diagnostic biomarker for BC, as our findings showed a significant downregulation in TNBC compared to luminal A. Furthermore, we also suggest from our study that sample specimens of such heterogenous malignancies should be collected using a more prudent technique such as LCM technology, so as to limit the diversity of tissue specimen and precisely select the desired area of interest from samples.


Our study had some limitations that must be acknowledged, which arose primarily from the relatively low sample size and unmatched characteristics of the compared groups; therefore, validation in large-scaled powered studies is warranted. Secondly, our study design did not include a comparison between fixed and non-fixed tissue sample to test the limitations of the FFPE extraction step. However, the available literature reports comparing paired fresh and FFPE tissues show that even though RNA integrity can decrease during FFPE extraction, reproducible and highly correlated gene expression data can be obtained from FFPE samples of various tissue types, including breast cancer subtypes [32,33,34,35]. In addition, the use of more than one representative cell line for each breast cancer subtype would have been ideal.

Although we did prove that some lncRNAs possess tissue-specificity in matters of BC and that they are able to significantly distinguish between tissue subtypes, future studies of larger cohort sizes are needed to confirm our findings with higher confidence.


## 5. Conclusions

BC still remains a public health concern due to its complexity and heterogeneity. LncRNAs show great potential in deciphering the molecular mechanisms of this malignancy. Moreover, it is suggested that lncRNAs could be used as diagnostic and prognostic biomarkers for breast carcinoma because of their differential expression in the various tissue subtypes. Our study showed that the expression level of GAS5 is significantly downregulated in TNBC tissues and cell lines compared to Luminal A, suggesting a promising potential role as a tissue-specific diagnostic biomarker for BC. Nevertheless, more in-depth studies with larger sample sizes are required in order to better investigate lncRNAs tissue specificity, as well as their molecular mechanisms in the biology of BC.

## Figures and Tables

**Figure 1 medicina-57-00371-f001:**
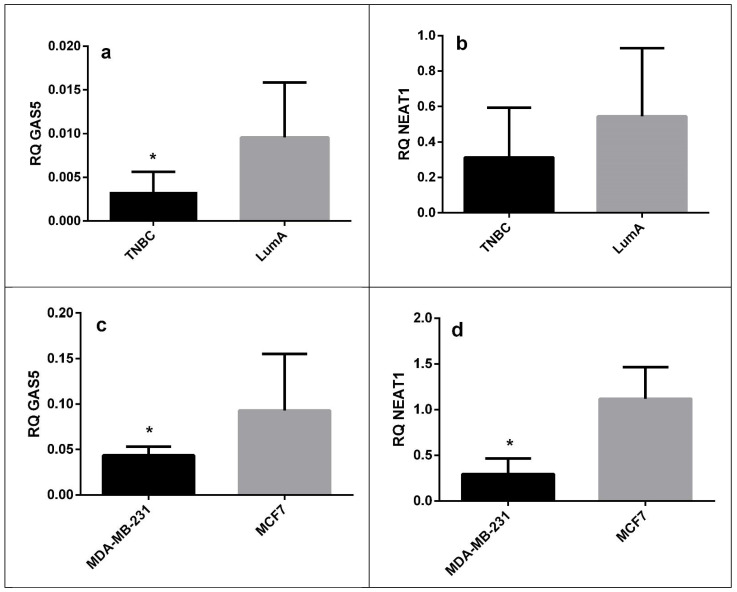
Relative quantity (RQ) of GAS5 (**a**) and NEAT1 (**b**) in the TNBC and Luminal A formalin-fixed paraffin-embedded (FFPE) samples and cell lines (**c**,**d**); * denotes *p* < 0.05.

**Table 1 medicina-57-00371-t001:** Patients’ characteristics and pathological tumor features.

Characteristics		TNBC *N* (%)	Luminal A *N* (%)
Age	>50	7 (77.78)	8 (88.89)
	<50	2 (22.22)	1 (11.11)
BMI	Normal	5 (55.56)	5 (55.56)
	Obese	4 (44.44)	4 (44.44)
Stage	I and II	5 (55.56)	4 (44.44)
	III	4 (44.44)	5 (55.56)
Tumor size	<5 cm	5 (55.56)	6 (66.67)
	>5 cm	4 (44.44)	3 (33.33)
Lymph node involvement	Yes	5 (55.56)	4 (44.44) *
	No	4 (44.44)	3 (33.33) *
HER2	Positive	5 (55.56)	5 (55.56)
	Negative	4 (44.44)	4 (44.44)
Ki-67	Positive	9 (100.00)	9 (100.00)
	Negative	0 (0.00)	0 (0.00)
ER	Positive	0 (0.00)	9 (100.00)
	Negative	9 (100.00)	0 (0.00)
PR	Positive	0 (0.00)	9 (100.00)
	Negative	9 (100.00)	0 (0.00)

* Percentages calculated from the total samples does not add up to 100% due to missing values (undetermined lymph node involvement for two patients in the Luminal A group).

**Table 2 medicina-57-00371-t002:** Differential expression of lncRNAs in the profiling study.

LncRNA Name	Fold Change	*p* Value	LncRNA Name	Fold Change	*p* Value
BCYRN1	0.967	0.457	MEG3	5.394	0.605
BDNF-AS	0.894	0.469	NEAT1	5.818	0.548
DISC2	0.624	0.325	OTX2-AS1	205.222	0.029
EGOT	1.634	0.860	PANDAR	1.101	0.236
FTX	13.518	0.456	PTENP1-AS	0.989	0.708
GACAT1	1.347	0.728	SNHG16	1.115	0.614
GAS5	1.734	0.914	SOX2-OT	0.020	0.042
H19	1.163	0.764	ST7-AS2	2.206	0.956
HEIH	1.388	0.702	TERC	1.652	0.812
HOTAIR	5.771	0.543	TMEM161B-AS1	0.909	0.474
IPW	1.163	0.656	TRERNA1	0.444	0.372
JPX	1.014	0.512	TUG1	1.453	0.814
KCNQ1OT1	4.028	0.690	UCA1	0.386	0.601
FALEC	1.204	0.633	XIST	0.769	0.316
LINC-ROR	0.624	0.174	ZFAS1	1.680	0.868
MALAT1	1.763	0.936			

**Table 3 medicina-57-00371-t003:** Differential expression of long non-coding RNAs (lncRNAs) in triple negative breast cancer (TNBC) versus Luminal A samples in the validation study.

LncRNA	Sample Type	Fold Change	* p * Value
GAS5	FFPE tissue	0.33	0.022
	Cell lines	0.52	0.035
NEAT1	FFPE tissue	0.66	0.27
	Cell lines	0.23	0.035

## Data Availability

The data supporting the findings are available from the corresponding author, upon request.

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
