# Peer review of "Long Non-Coding RNA Expression in Laser Micro-Dissected Luminal A and Triple Negative Breast Cancer Tissue Samples—A Pilot Study"

_medicina, 2021, doi:10.3390/medicina57040371_

Round 1

Reviewer 1 Report

publication can be accepted for publication

Author Response

We thank the reviewer for the acceptance of the manuscript. 

Reviewer 2 Report

The Manuscript “ Long non-coding RNA expression in laser Micro-dissected LuminaA and Triple Negative Breast Cancer Tissue Sample – a pilot study” Anca Marcu et al., performed lncRNA profiling in 18 breast cancer patients for total four LncRNA (OTX2-ASI and SOX2OT, NEAT1 and GAS5) also explained the importance of laser dissection in lncRNA profiling of the select region from the tissue. However, the manuscripts have serious flaws and lack clarity as mentioned below.

Abstract:

  • Lacks clarity, how many of them are Triple negative and Luminal A in these 18BC samples?
  • What exactly does authors mean when they say differentially expressed in the profiling but did not reach the statistically significant in individual validation measurement? What kind of statistically validation? Needs a better explanation.
  • Why only GAS5 and NEAT1 were picked for the study, no proper explanation.
  • “NEAT1 was significantly downregulated only in TNBC cell vs Lumina A” is Lumina A cell line used?

Materials and Methods:

 “A fixed volume of 8 ul from the total RNA was input” is all the 18 samples quantified for total RNA? 8ul is your cDNA used for RTqPCR or RNA converted to cDNA? Very confusing statement

Results:

  • Were all the genes in the Table checked in 18 BC samples?
  • Primers information not provided
  • Figure not sound. Significance score is not used. Y-axis is a relative expression or folds change?
  • At least two cell lines to be used for TNBC (for ex: MDA-MB231, MDA-MB468) and Lumina A (MCF7 and T47D).
  • How many TNBC and Luminal A samples in 18 BC FFPE?
  • You can refer to CCLE, GTEX, and TCGA databases

Language needs to be checked throughout and have more clarity.

Author Response

We thank the reviewers for their insightful observations. We have revised the manuscript according to the reviewers’ suggestions, as presented in the below point by point response.

Reviewer 2

The Manuscript “Long non-coding RNA expression in laser Micro-dissected LuminalA and Triple Negative Breast Cancer Tissue Sample – a pilot study” Anca Marcu et al., performed lncRNA profiling in 18 breast cancer patients for total four LncRNA (OTX2-ASI and SOX2OT, NEAT1 and GAS5) also explained the importance of laser dissection in lncRNA profiling of the select region from the tissue. However, the manuscripts have serious flaws and lack clarity as mentioned below.

Abstract:

  • Lacks clarity, how many of them are Triple negative and Luminal A in these 18BC samples?

Response: We added an explanation (lines 27-28). Detailed explanation can be found in Table 1.

  • What exactly does authors mean when they say differentially expressed in the profiling but did not reach the statistically significant in individual validation measurement? What kind of statistically validation? Needs a better explanation.

Response: We clarified in the abstract that first there was a profiling experiment done by PCR array, followed by individual validation using individual assays. So when using specific individual assays, OTX2-AS1 and SOX2OT did not reach statistical significance.

  • Why only GAS5 and NEAT1 were picked for the study, no proper explanation.

Response: It is explained in the Abstract (lines 33-34) Results (193-195) and an extra explanation can be found in the Discussion section (228-229), supported by references 22-26.

  • “NEAT1 was significantly downregulated only in TNBC cell vs Lumina A” is Lumina A cell line used?

Response: Yes, detailed in the Methodology section (2.6.), and also added in the abstract that cell lines were used in addition to tissue samples for validation.

Materials and Methods:

 “A fixed volume of 8 ul from the total RNA was input” is all the 18 samples quantified for total RNA? 8ul is your cDNA used for RTqPCR or RNA converted to cDNA? Very confusing statement

Response: Yes, each sample was treated the same, meaning that 8ul from the total RNA extracted was used for cDNA conversion by reverse transcription, as specified in the manufacturer’s protocol.

Results:

  • Were all the genes in the Table checked in 18 BC samples?

Response: Yes, as indicated in the methods section, the RT2 lncRNA PCR Array Human lncFinder (LAHS-001ZE; Qiagen, Germany) panel analyses 84 lncRNA genes in the profiling step.

  • Primers information not provided

Response: Unfortunately, we are not able to provide the primer and probes details, because these are proprietary information of the manufacturer (Thermo Fisher) and are not available. However, we added the assay IDs that were used. (section 2.5 from Materials and Methods, lines 151-152).

  • Figure not sound. Significance score is not used. Y-axis is a relative expression or folds change?

Response: We added in the legend of figure 1 that RQ means relative quantity for the Y-axis. We also added the p values for each of the graph.

  • At least two cell lines to be used for TNBC (for ex: MDA-MB231, MDA-MB468) and Luminal A (MCF7 and T47D).

Response: We agree with the reviewer that additional cell lines would add credibility to our data. However, as stated in the methods section, we only used one cell line for each breast cancer subtype: MDA-MB231 for TNBC and MCF7 for Luminal A. Unfortunately, we do not possess the other two cell lines suggested by the reviewer and certainly we cannot purchase them and do the additional experiments at this point. We added this limitation of our study in the discussion section.  

  • How many TNBC and Luminal A samples in 18 BC FFPE?

Response: 9 TNBC and 9 Luminal A. We explained in the Abstract (lines 27-28) and detailed in Table 1.

  • You can refer to CCLE, GTEX, and TCGA databases

Response: This is a pilot study with the objective of assessing that lncRNA expression data can be obtained from FFPE tissue samples. The second exploratory objective was to see if there are any lncRNAs that are differentially expressed between the two breast cancer subtypes investigated.

Round 2

Reviewer 2 Report

Thank you for the revision. Authors need to be careful when you write for ex, line 196 " downregulated in both TNBC tissue sample and celline compared to Luminal A"  TNBC tissue sample and cell line compared to Luminal A (celline or tissue)?. authors needs to be very specific. These confusion throughout the manuscript needs to be corrected.

Figure needs to be modified. When authors present with significant value, it is necessary to use asterisks to that you are comparing with.  figure has to be uniform with the same font and style. Please mention a,b,c,d in the figure.

Author Response

Thank you for the suggestions. Here you have our response.

This manuscript is a resubmission of an earlier submission. The following is a list of the peer review reports and author responses from that submission.

Round 1

Reviewer 1 Report

In this manuscript, Anca Marcu et al., studied Lnc RNA Expression in Laser Micro-dissected Luminal A and Triple Negative Breast Cancer Tissue Samples. In this Study, Author performed  using lncRNA profiling in TNBC and luminal A tissue samples from 18 Breast cancer patients. sample size is very small with that they found GAS5 showed  expression level difference in TNBC tissues and compared to luminal A. Based on the results with smaller sample size author try to convince that LCM could be a useful technique for gene expression analysis.

It will be helpful if the authors can  provide primer details of GAS5 and NEAT1in the methodology section

In general, I find the manuscript very clearly written and over all  this is a good paper.

Author Response

We thank the reviewers for their insightful observations. We have revised the manuscript according to the reviewers’ suggestions, as presented in the below point by point response.

Response: Unfortunately, we are not able to provide the primer and probes details, because these are proprietary information of the manufacturer (Thermo Fisher) and are not available. However, we added the assay IDs that were used. (section 2.5 from Materials and Methods, lines 151-152).

Reviewer 2 Report

The goal of this paper was to analyze differential expression of lncRNA from different BC subtypes.  The authors used LCM to isolate specific groups of malignant cells from archived FFPE tumor tissue slides and cell lines.

While this paper had great promise, there were many flaws both in the design and the analysis steps that preclude its publication at this time.

Design:

Since the authors are using archived FFPE samples they needed to have controls to show the quality of their transcriptomics analyses as well as the difference in the profiles one would get from FFPE samples compared to non-FFPE samples.

First, the authors needed to run parallel experiments on fresh (non FFPE) samples vs FFPE samples in order to determine what the effects of FFPE and deparaffinization had on the downstream transcriptomics analyses.  One would expect some major differences in the RNA quality/quantity obtained due to the sample preparations, but the authors did not show this data.  Only if the authors can show that there is no difference in the downstream transcriptomics analyses can they try to compare FFPE to non-FFPE samples (which is what they did by comparing the archived FFPE tumor tissue to cell lines).  This is not a valid comparison.  The authors must test for expression differences between fixed and non fixed tissue samples to uncover whether FFPE extraction step is not limiting.

The authors should have used archived FFPE samples from patient that had non-malignant tumors to directly compared them to the malignant tumor slides.  All samples should have undergone the same processes (I.e., LCM isolation, RNA extraction etc…).

One of the big claims of this paper was the fact that the authors used LCM to isolate the malignant cells of interest.  Yet there is no description of the methods at all.  Beside the name of the instrument used, there is nothing of value in the M&M section.  Nobody can use that info to try to replicate these results.

It is especially important since the author state on page 6 (l. 187-188)” “To our knowledge, this is the first study to assess the differential expression of GAS5 lncRNA in different BC tissue subtypes using LCM technology.”  

The same is true for the deparaffinization step.  A brief, one sentence, of how this was done should have been included so that others can determine the effect that this step might have on the samples and for reproducibility purposes.

The authors did not followed the Authors Guidelines for the Materials and Methods section.

Basically, section 2.2 and 2.3 should have been explained the way section 2.4 was. 

Analysis:

First, a power analysis in this type of study is essential to be sure that the statistical analysis is valid (especially with p values so close to the cutoff).

Next, a key assumption of the 2-ΔΔCT method is that the amplification efficiencies of the target and reference must be approximately equal.  No data is provided on amplicon efficiency therefore the underlying data is hard to judge.  This is important since 2-ΔΔCT cannot be used if this assumption is not met.

Patient characteristics between TNBC and Luminal A do not seem to be well matched, therefore the comparative data may not be providing evidence of differences between cancer types but instead groups.  A statistical power analysis would be helpful here.

The authors need to be more clear about what exactly is the “screening step” versus the “profiling step” versus the “validation step.”

The conclusions in the paper are extremely mixed.  GAS5 and NEAT1 are not statistically significant at the LCM screening stage, yet are validated anyway (where they are suddenly significant).  Then the two IncRNA that are identified via the LCM method are not significant in the validation step.  Therefore what was the purpose of the LCM step if the RNA identified by it were not validated (and the only RNA that were validated were not identified by the LCM method.)

Author Response

We thank the reviewers for their insightful observations. We have revised the manuscript according to the reviewers’ suggestions, as presented in the below point by point response.

Round 2

Reviewer 2 Report

I appreciate the difficulties in trying to set up a pilot study.  However, a pilot study doesn’t mean that you do not need the proper controls etc…. A pilot study is a “small study so that protocols, methods, strategies etc…” can be tested.  It is there to help larger scale research and there to identify possible problems etc…. This is not what was done in this case.  The authors did add some info in the M&M and basically, agreed with most of the point raised during the first review, but simply stated that they can’t change anything because this is a pilot study…. These are not valid answers. Here are some examples:   1) “Ideally, this should have been be the case. However, these FFPE samples were obtained from the hospital archive and we were not able to get matched non-FFPE samples for comparisons.”   2) "It is true that lncRNA quantity and quality is affected by FFPE extraction, but previous literature reports show that in general, reliable gene expression data can be obtained from FFPE samples, highly correlated with fresh samples in various tissue types, including breast.”  How can it be true that “the quality and quantity of FFPE samples is affected”…… yet data from FFPE “highly correlated with fresh samples in various tissue types, including breast.”  The authors need to show that this is the case here, in their hands, with their methods.   3) “Unfortunately, we did not have access to non-malignant tumor FFPE tissues. Moreover, as these kind of comparisons were done previously, we intended to focus on the differences between the two breast cancer subtypes that have different prognosis." Again, since the authors are doing a “pilot study” and want to show the usefulness of LCM in comparing different breast cancer tissue, they can’t just say “these kind of comparisons were done previously.”  They were probably not done with the conditions proposed in this study and therefore, even if the authors want to focus on the differences between the 2 breast cancer subtypes, they still need to demonstrate the usefulness of their technique and need the proper controls with ALL samples having undergone the same processes, which is not the case here.   4) "We agree with the reviewer about the assumption regarding PCR efficiency; however, this assumption is always implied in published papers regarding gene expression data.”  This is not a valid answer.   5) “We agree with the reviewer and added this as a limitation to our study in the discussion.”  Simply stating that it is a problem in the discussion doesn’t answer the question.     6) “All FFPE samples analysed in both stages (profiling and validation) were LCM cut, RNA extracted and lncRNA expression analysed. The goal of the profiling step was to identify statistically significant lncRNAs, which would eventually be validated. This was not the case, most likely due to power limitations that we acknowledged. The first two lncRNAs that were statistically significant in the profiling step were not further validated (did not reach statistical significance in the individual validation assay using specific primers). Therefore, we selected GAS5 and NEAT1 from literature for validation, as they are two of the most well studied lncRNAs associated with BC, as we explained in the manuscript and cited references [22-26].   This was exactly the point that was raised in the review.  The authors missed their 1st goal which was to identified statistically significant lncRNAs.  The two that were statistically significant in the profiling step were not statistical significant. in the validation.  Doesn’t these results show that this pilot study needs to be refined and as is right now couldn’t really be used?  Hence, the authors had to look at GAS5 and NEAT1, not based on their data but based on literature validation.   As the authors have shown themselves, at this stage, there are way too many limitations to this pilot study and it needs to be reassessed and further developed to show that it can be successfully used in the future.

Author Response

We have responded to the reviewer's comments in the attachment below
